# Deformation Characteristics and Control Method of Kilometer-Depth Roadways in a Nickel Mine: A Case Study

**Guang Li** [1,2,3], **Fengshan Ma** [1,2,*], **Jie Guo** [1,2] **and Haijun Zhao** [1,2]

1    Key Laboratory of Shale Gas and Geoengineering, Institute of Geology and Geophysics,
     Chinese Academy of Sciences, Beijing 100029, China; liguang@mail.iggcas.ac.cn (G.L.);
     guojie@mail.iggcas.ac.cn (J.G.); zhaohaijun@mail.iggcas.ac.cn (H.Z.)
2    Innovation Academy for Earth Science, CAS, Beijing 100029, China
3    University of Chinese Academy of Sciences, Beijing 100049, China
*    Correspondence: fsma@mail.iggcas.ac.cn

**Abstract:** Deformation failure and support methods of roadways have always been critical issues in mining production and safety, especially for roadways buried in complex engineering geological conditions. To resolve these support issues of kilometer-depth roadways under high ground stress and broken rock mass, a case study on the roadways in the No. 2 mining area of Jinchuan Mine, China, is presented in this paper. Based on a detailed field survey, the deformation characteristics of the roadways and failure modes of supporting structures were investigated. It was found that the horizontal deformations were serious, and the primary support was not able to control the surrounding rock well. Additionally, a broken rock zone test was carried out, which indicated that a zonal disintegration phenomenon occurred around the roadways and the maximum depth of the fractured zone was more than 4.8 m. In order to effectively limit the deformation in the roadways, a new support scheme called the "multistage anchorage + concrete-filled steel tube" was put forward. To further assess the support behavior of the new method, we selected a test roadway in the research area, and numerical simulations and in-situ monitoring were conducted. The findings suggest that the roadway's serious deformation under high ground stress and broken rock mass could be successfully controlled by the new control method, which can provide a reference for other engineering solutions under similar geological conditions.

**Keywords:** 1000 m-depth roadway; deformation characteristics; zonal disintegration; FLAC$^{3D}$; concrete-filled steel tube

## 1. Introduction

   Due to fast economic growth, the status of mineral resources in the national economy and the need for mineral resources has been increasing. In mining production, controlling roadway deformation is an important task for maintaining enduring and effective supports that directly affect safety and high-efficient production. Controlling roadway deformation using appropriate support measures has become a critically important focus area in the study of rock mechanics and mining [1].

   Recently, the deformation characteristics of roadways have been a hot research topic and have received considerable attention. Coggan et al. [2] and Huang et al. [3] investigated the effect of a soft interlayer on the deformation of roadways. They found that the weak interlayer had an important effect on the degree of failure, causing asymmetrical stress. Tu et al. [4] presented a mechanical roadway model based on a theoretical analysis and predicted the stress distribution around the roadway. Yan et al. [5] explored the fault's influence on the deformation mechanism of a roadway and found

that roadway convergence increased with activation of the fault. Furthermore, Yu et al. [6] studied the deformation laws in a goaf heading face based on a numerical simulation and field survey method. Chang et al. [7] and Gao et al. [8] discussed the characteristics of roadway deformations. They pointed out that high ground stress, mining-induced stress and low strength rock cause large deformation of roadways. Chen et al. [9], Kang et al. [10] and Yang et al. [11] used numerical simulations to investigate a deep roadway's deformation failure characteristics, including Universal Distinct Element Code (UDEC), Fast Lagrangian Analysis of Continua in 3 Dimensions (FLAC$^{3D}$) and LDEAS 1.0. Moreover, Shen [12] divided roadway failures into roof sags, rock falls, beam failures, rib failures, skin failures and shearing failures.

Meanwhile, traditional supporting methods, such as the use of spray-bolt-mesh, can no longer effectively control the large deformation of roadways under high geo-stress and broken rock mass. Researchers are thus paying more attention to the development of new materials and combined support schemes.

He et al. [13] and Sun et al. [14] developed bolts with a constant resistance; Li et al. [15] found a high-strength cable; Srivastava and Singh [16] developed a grouted bolt; and Gao et al. [17], Li et al. [18] and Zhang et al. [19] proposed round, square, and D-shaped concrete-filled steel tubes. Furthermore, Aksoy et al. [20] put forward a non-deformable support system (NDSS). Shen et al. [21] experimentally analyzed the characteristics of five concrete-filled steel tube arches used in deep roadways based on theory and numerical simulation. Additionally, Yang et al. [22] presented a support method, including double-layer-truss and bolt-mesh-cable, which can effectively control rheological deformation. Guo et al. [1] designed a coupled truss support scheme including anchors, anchor bolts, and shotcrete. Moreover, Meng et al. [23] used U-shaped steel to limit the floor heave. Wang et al. [24] put forward a novel reinforcement technology (whole section anchor-grouting) for roadways in rock mass with fractures. Lv et al. [25] adopted square steel confined concrete to control surrounding rocks with high stress. Additionally, Lu et al. [26] put forward a highly resistant and yielding support system for long-term control. Barla et al. [27] provided a hybrid system with yielding steel ribs and Yang et al. [11] developed a hybrid system with bolt, cable, mesh, shotcrete and shell.

Although many researchers have studied the deformation and support of roadways which has resulted in a series of achievements, most of the solutions are based on coal mines and many support methods that cannot be widely used because of their high cost. Few studies have been conducted in large metal mines, especially those with a kilometer depth characterized by broken rock mass and high in-situ stresses.

Therefore, the Jinchuan nickel mine in China was selected as the research area in this study and a detailed field investigation into the deformation and support of deep roadways under broken rock mass was performed. We then conducted a broken rock zone test in this area to assess the primary control method's supporting effect. Moreover, to compensate for its disadvantages, a new hybrid control system with multistage anchorage and a concrete-filled steel tube (CFST) was designed. Numerical calculations using the Fast Lagrangian Analysis of Continua in 3 Dimensions (FLAC$^{3D}$) were performed for a simulation and comparison of the evolutionary processes of the deformation, as well as the stress of roadways and support structures. Additionally, on-site experiments, field monitoring and economic analyses were performed to assess the new support scheme's control effect.

## 2. Geological Environment and Engineering Properties

As China's biggest nickel production base, Jinchuan Mine is situated in Jinchang City, Gansu Province, as shown in Figure 1. It consists of four mining zones, where the No. 2 mining area has the largest ore body, with an overall length of approximately 3000 m, a surface outcrop length of about 2500 m, a mean breadth of 380 m, and a downward extension of over 1000 m. The No. 2 mining area is situated in the desert at the southern edge of the Chaoshui basin, surrounded by the Longshou mountains in the southwest, as shown in Figure 2. The terrain is rugged, with an average altitude of 1750 m, so the influence of the arid climate and simple hydrogeological conditions is negligible.

This mining area has a rather complicated history with regard to the geological background. It has experienced many geological tectonic activities, such as Indosinian movement, Yanshan movement and Lvliang movement. Many kinds of magma activities are also frequent in this area. The mining area's main structure consists of monoclines with a southwest tendency, and between the layers folds propagate, frequently forming small anticlines and synclines. These difficult geological conditions, crisscrossed structural surface, weak rock strength and high ground stress all hinder the stability of the rock surrounding the roadway.

In the No. 2 mining area, the current state of the on-site stress is compressive stress, with horizontal stress being the maximum principal stress. In the shallow part, within a depth of about 30 m from the surface, the maximum principal compressive stress is approximately 3 MPa. When the depth increases, the stress value increases. The maximum principal stress reaches 30 MPa and the highest value is 52.2 MPa between a depth of 400 and 800 m. Its direction is mostly NE–NNE, practically vertical to the ore body's orientation, and there is a large difference between the minimum and maximum principal stress [28–30].

A downward filling mining method with a subsection height of 20 m is adopted in the Jinchuan Mine. Each subsection is further divided into five layers with a height of 4 m and the ore body buried at 900 m is the main mining area at present. Trackless transport is used in the stope, so it is necessary to construct transport roadways with large sections. The main object of this study is the transport roadway, which is often a permanent project and needs to provide long-term stable support.

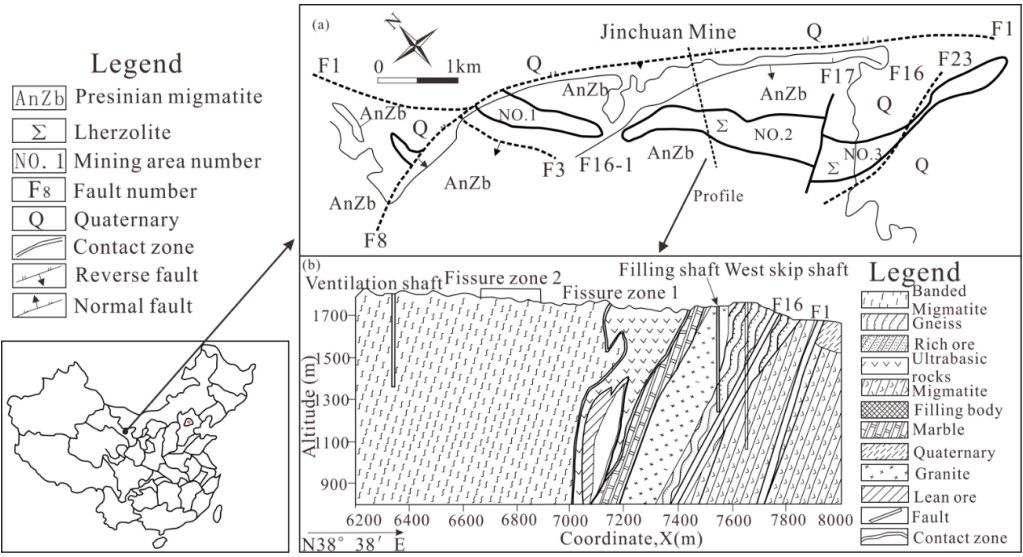

**Figure 1.** Engineering geological background of the Jinchuan Mine. (**a**) Jinchuan Mine's location and geology. (**b**) Jinchuan No.2 Mine's geological section along an exploratory line.

## 3. Roadway Deformation Characteristics

### 3.1. Field Test

Conducting a field investigation is a suitable way to understand the deformation failure characteristics of roadways visually and in detail. In this study, it was performed on the Jinchuan No. 2 Mine's panel (814 m above sea level with a burial depth of approximately 1000 m). To provide tangible data for the adoption of appropriate support measures, the investigation included the rock mass strength, support patterns, ground stress, construction methods and other factors that impact the stability of deep roadways.

There are five main lithology types in the survey area, and Table 1 shows the mechanical and physical parameters of these intact rocks and rock masses. It can be seen that the intact rock has a high

strength compared to that of the rock mass. Therefore, the roadway is very weak. Furthermore, the ultrabasic rock mass has better engineering geological conditions than the marble rock mass.

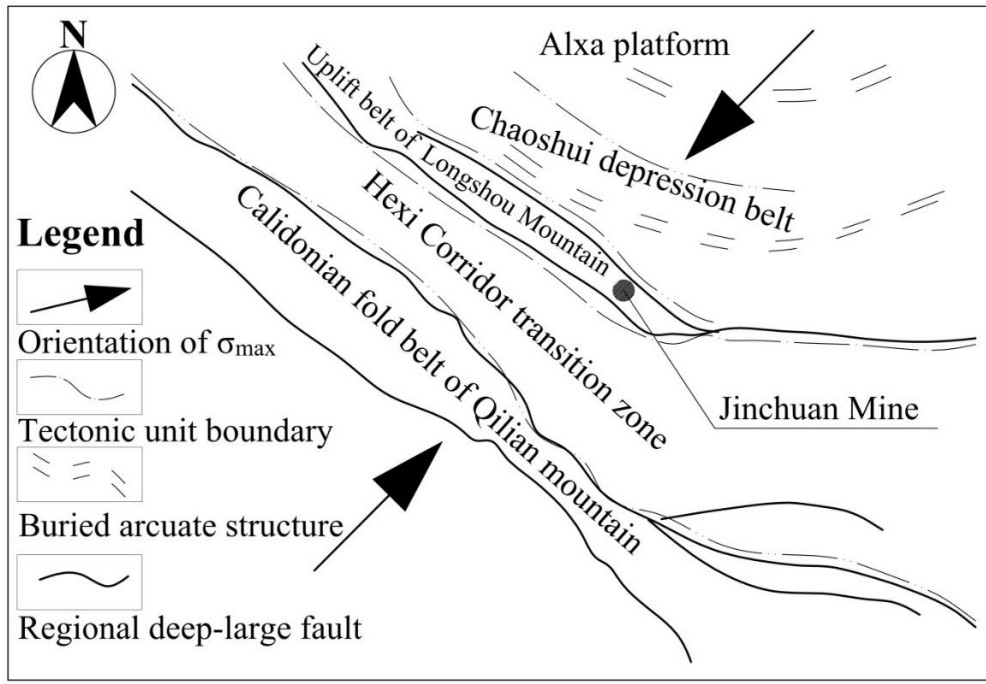

**Figure 2.** Tectonic map of Jinchuan.

Wu et al. [31] studied the geo-stress in Jinchuan mining for several years, based on a hollow inclusion strain gauge. They measured six points in the study area, and Table 2 displays the stress parameters of each point. It is illustrated that the maximum principal stress reaches 46.6 MPa and the gap between the minimum and maximum principal stress is huge, which indicates a high shear stress.

**Table 1.** Mechanical and physical parameters of rocks and rock masses.

| Lithology | Type | Density (g·cm$^{-3}$) | Tensile Strength (MPa) | Compressive Strength (MPa) | Cohesion (MPa) | Internal Friction Angle (°) | Elastic Moduli (GPa) | Poisson's Ratio | RMR |
|---|---|---|---|---|---|---|---|---|---|
| Marble | Intact rock | 2.86 | 11.00 | 104.0 | 9.5 | 42.0 | 64.0 | 0.16 | - |
| | Rock mass | 2.40 | 0.80 | 15.0 | 0.7 | 35.0 | 6.0 | 0.26 | 39.56 |
| Migmatite | Intact rock | 2.73 | 9.09 | 51.9 | 15.5 | 32.0 | 55.0 | 0.15 | - |
| | Rock mass | 2.45 | 1.00 | 17.0 | 0.8 | 35.0 | 6.4 | 0.25 | 40.27 |
| Ultrabasic rock | Lherzolite | 2.93 | 13.00 | 150.0 | 11.5 | 43.5 | 95.0 | 0.23 | - |
| | Rich ore | 3.02 | 9.60 | 73.5 | 10.0 | 45.0 | 70.0 | 0.22 | - |
| | Poor ore | 3.05 | 6.60 | 70.0 | 7.0 | 43.5 | 65.0 | 0.31 | - |
| | Rock mass | 2.50 | 2.20 | 25.0 | 0.8 | 38.0 | 7.0 | 0.24 | 43.23 |

**Table 2.** Ground stress parameters.

| No. | Lithology | $\sigma_1$ | | | $\sigma_2$ | | | $\sigma_3$ | | |
|---|---|---|---|---|---|---|---|---|---|---|
| | | Value (MPa) | Direction (°) | Dip Angle (°) | Value (MPa) | Direction (°) | Dip Angle (°) | Value (MPa) | Direction (°) | Dip Angle (°) |
| G1 | Marble | 42.5 | 43 | 43 | 15.8 | 98 | −27 | 11.3 | 171 | −28 |
| G2 | Ultrabasic rock | 43.8 | 28 | −21 | 16.7 | 131 | −28 | 12.0 | 87 | 53 |
| G3 | Marble | 46.6 | 193 | 65 | 25.3 | 89 | 7 | 19.1 | 176 | −25 |
| G4 | Marble | 44.4 | 224 | 21 | 31.8 | −45 | 3 | 24.2 | 232 | −68 |

The test roadway's initial support was a compound support scheme. It consisted of a bolt-mesh-anchor (double layer) and U-shaped steel, as depicted in Figure 3. The cement mortar anchor rod was constructed with secondary rebar (spacing, 1000 × 1000 mm; length, 2250 mm; diameter, 22 mm). An ordinary steel plate (thickness,10 mm; size, 200 × 200 mm) was utilized to make an anchor

plate. Concrete (25 MPa compressive strength) was used for the shotcrete lining, which had a 100 mm thick single layer. U36 steel was used for the U-shaped steel support [32].

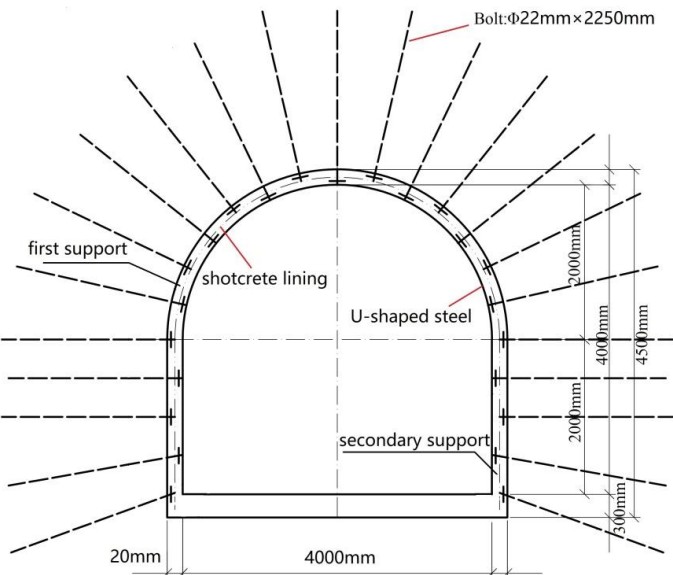

**Figure 3.** Primary support method used in the test roadway.

Moreover, a judgment criterion, which is named the LG criterion and is used for determining the degree of roadway deformation under the supporting structure is proposed in this paper. If the axial deformation exceeds the distance of the supporting structure by two times or the radial deformation exceeds the maximum elongation of the supporting structure, it is considered severe deformation. The spacing of the supporting structure in this study area is 1 m and the maximum elongation of the bolt is 45 cm. Therefore, a deformation failure point with an axial deformation exceeding 200 cm or radial deformation exceeding 45 cm is defined as severe, based on the LG criterion. Roadways that were about 700 m long were examined and 48 deformation points were found, as shown in Figure 4.

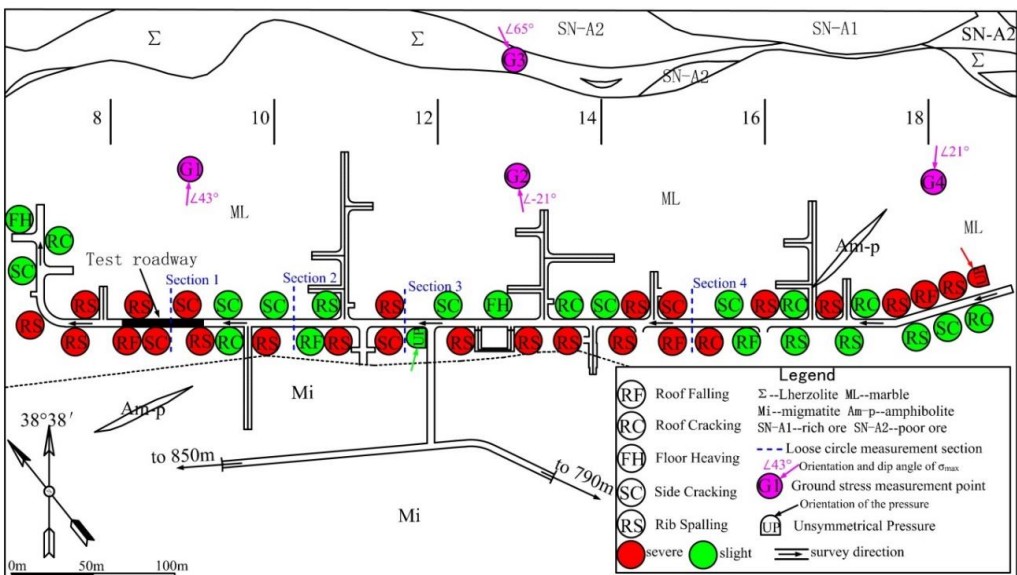

**Figure 4.** Field investigation of the panel at the burial depth of approximately 1000 m.

## 3.2. Deformation Characteristics of Roadways

Figure 5 shows that the roadway deformation failure modes in this area can be sorted into six groups including roof falling, roof cracking, unsymmetrical pressure, floor heaving, side cracking and rib spalling. Additionally, each mode's occurrence time and the total percentage were calculated, as illustrated in Figure 6 [33].

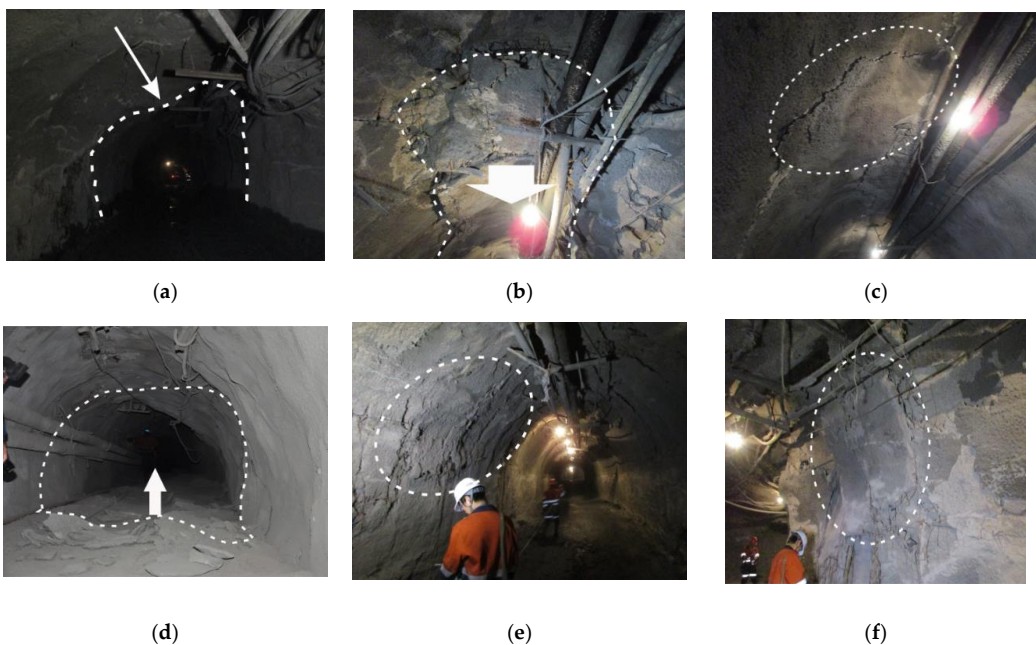

(**a**)        (**b**)        (**c**)

(**d**)        (**e**)        (**f**)

**Figure 5.** Deformation failure modes: (**a**) Unsymmetrical pressure, (**b**) Roof falling, (**c**) Roof cracking, (**d**) Floor heaving, (**e**) Side cracking, (**f**) Rib spalling.

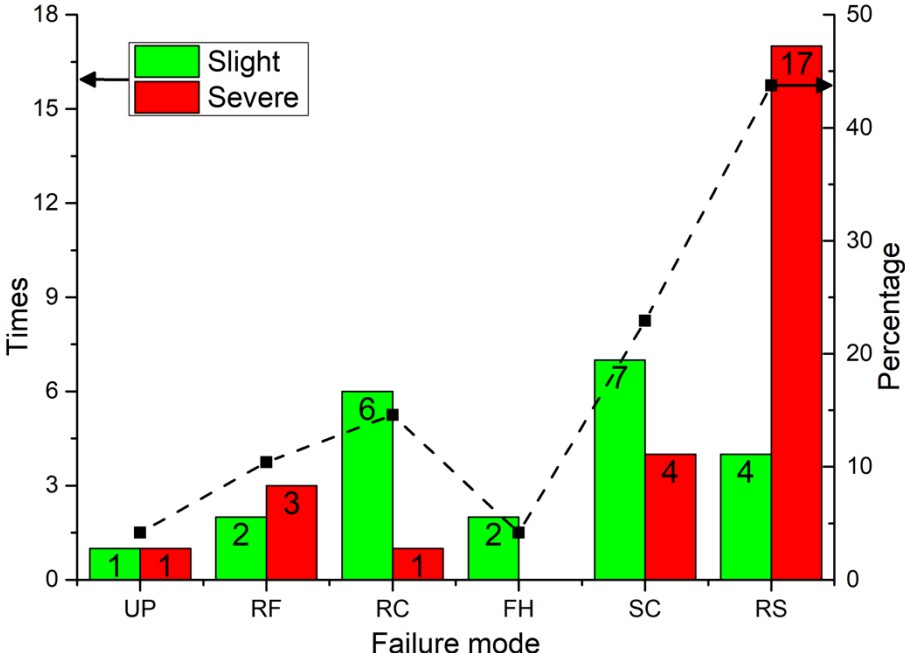

**Figure 6.** Times and percentage of various failure modes.

Based on the figures above, we conducted comprehensive analyses of the field investigation results, and can conclude the following with regard to the roadway's deformation failure characteristics.

### 3.2.1. Maximum Principal Stress Plays a Dominant Role

In the research area, the ground stress is very high, while the horizontal stress is three times higher than the vertical stress. Figure 6 shows that the roof cracking, rib spalling and side cracking which are all involved in horizontal stress, account for more than 80% of the total deformation points. When the roadway is under a horizontal squeezing force, the weak structural surfaces in the original rock mass move towards the free face and tension fissures occur if this force surpasses the resistance strength of the supporting body. In particular, as the tunnel direction is vertical to the direction of the principal stress, the deformation of roadways becomes more serious. Moreover, we can see from Figure 4 that the unsymmetrical pressure's direction is almost accordant with the maximum principal stress's strike and there is very little roof falling since σ1 is almost horizontal.

### 3.2.2. Prominent Time Effects

The time effects of the roadway deformation include three aspects: a quick deformation rate in the initial stage, a high deformation speed and a lengthy deformation duration (Figure 7). When the roadway is excavated, unloading of the surrounding rock mass is fierce because of the intensive on-site stress. The deformation occurring during the initial stage is able to account for over 40% of the surrounding rock's entire deformation, although it only continues for a few weeks. Due to the high ground stress in the research area, the maximum deformation rate can reach more than 6 mm/d, according to the field measurement. Additionally, some roadways fail to achieve stability for a couple of years because the rock mass is generally soft, and the broken rock has significant rheology.

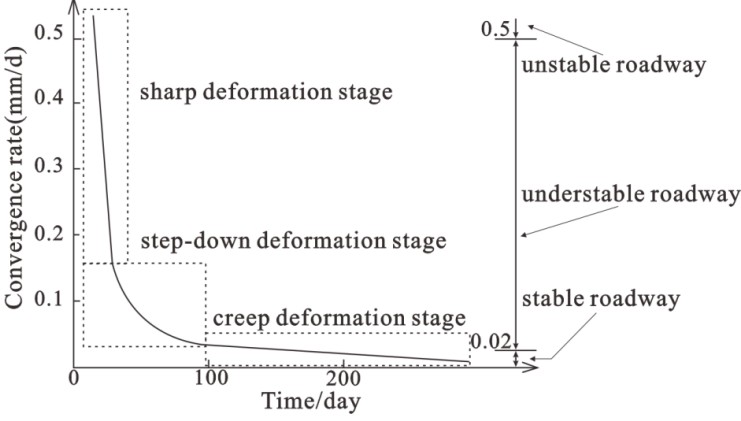

**Figure 7.** A typical convergence rate–time curve of studied roadways.

### 3.2.3. Significant Space Effects

The space effects of the roadway deformation include two aspects, universality and differences. Among the major systems in the study area, including transportation system engineering, segmented mining and quasi engineering, different chamber engineering, and so forth, practically all of the roadways have been deformed and destroyed. In order to meet the production needs, roadways are distributed in different areas, and the deformation and damage of these roadways show obvious differences. In terms of the same area, the roadway's convergence deformation in the ore body's upper wall is generally larger than that in the foot wall, and in terms of the same roadway, the amount of horizontal deformation is generally greater than the amount in the vertical direction due to the high horizontal ground stress value.

### 3.3. Failure Modes of Supporting Structures

Although a combined support is employed to prevent deformation, influenced by high ground stress and broken rock mass, it is still rather common to observe severe failure of the support structure.

In the field study, four support failure modes were detected: (1) bolt pulled out and slipped; (2) wire mesh twisted and snapped; (3) shotcrete cracked and extruded; and (4) U-shaped steel extruded and bent. These are depicted in Figure 8.

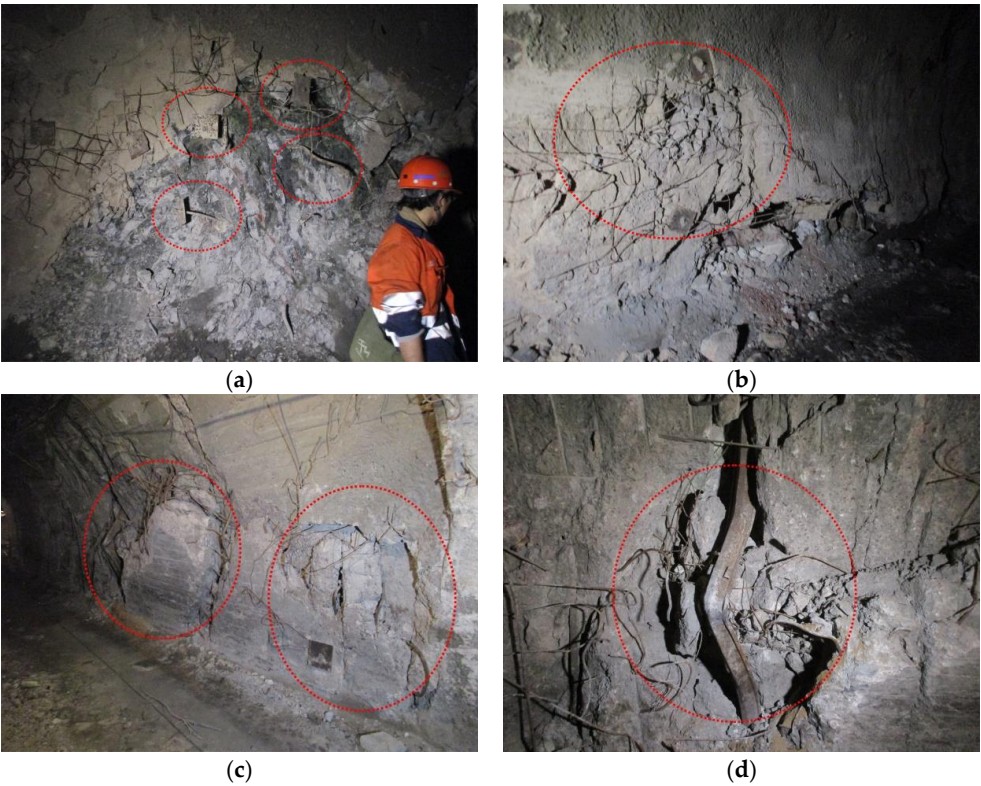

**Figure 8.** Support failure modes: (**a**) Bolt pulled out and slipped, (**b**) Wire mesh twisted and snapped, (**c**) Shotcrete cracked and extruded, (**d**) U-shaped steel extruded and bent.

Most of the failure bolts found in the investigation converged toward the air face of the roadway with the surrounding rock and only a small number were damaged on the steel plate and buried in the rock mas, which indicated that the bolts did not fully function in terms of supporting the surrounding rock. If the bolt length is greater than the fractured surrounding rock's depth, the end steel plate will be torn instead of being pulled out when the roadway has large convergence deformation. The shotcrete lining was broken and produced a cavity under the effects of severe rib spalling and sidewall squeezing. Based on the failure characteristics, the mechanical principle was shear failure or tension failure. The phenomenon of U-shaped steel being squeezed and bent often happened, especially on the rib spalling's failure point. It extruded and exposed the surrounding rock, and there was also an entire row of steel arches squeezed on the sidewall foot.

### 3.4. Rock Mass's Zonal Disintegration around the Roadways

Excavation of the roadway in the primary rock mass destroyed the original stress balance of the surrounding rock and redistributed the stress. On one hand, its radial stress decreased, and the tangential stress increased, which led to stress concentration. On the other hand, the surrounding rock stress state changed from triaxial to approximately biaxial, and the rock strength decreased. If the concentrated stress was less than the declined rock strength, the surrounding rock would carry itself and maintain an elastic state, that is, the support was unnecessary. Otherwise, the surrounding rock would gradually crack from the periphery to the inside, until it reached another new triaxial stress equilibrium state. At this time, an annular fractured zone occurred around the excavation space, which was the broken rock zone. Owing to the fact that the rock mass's mechanical properties were seriously

influenced by the stress environment, the rock with an elastic state in the shallow part displayed elastic–plastic or plastic deformation characteristics in the deep area, and the convergence deformation mechanism of the deep roadway was quite different from that of the shallow roadway. Hence, the distribution characteristics of the broken rock zone of the deep roadway were more complex, and accurately grasping the range of the broken rock zone was crucial for support selection.

Therefore, four sections in the study area were selected to conduct the broken rock zone test, as shown in Figure 4. Four measurement points were chosen for each section, two points on the spring line and two points on the spandrel. In order to improve the efficiency, a single-emission and double-receiving measurement method based on the ultrasonic wave was adopted, as shown in Figure 9. Waves emitted by the acoustic emission transducer T slid along the wall of the borehole, and they were then refracted back into the holes and received by the receiving transducers $R_1$ and $R_2$. The calculation formula of the rock mass's longitudinal wave velocity $V_p$ is as follows [34]:

$$V_p = \frac{\Delta L}{t_2 - t_1},$$ (1)

where $\Delta L$ is the distance between the two receiving transducers and $t_1$ and $t_2$ are the times when transducers $R_1$ and $R_2$ receive the wave, respectively.

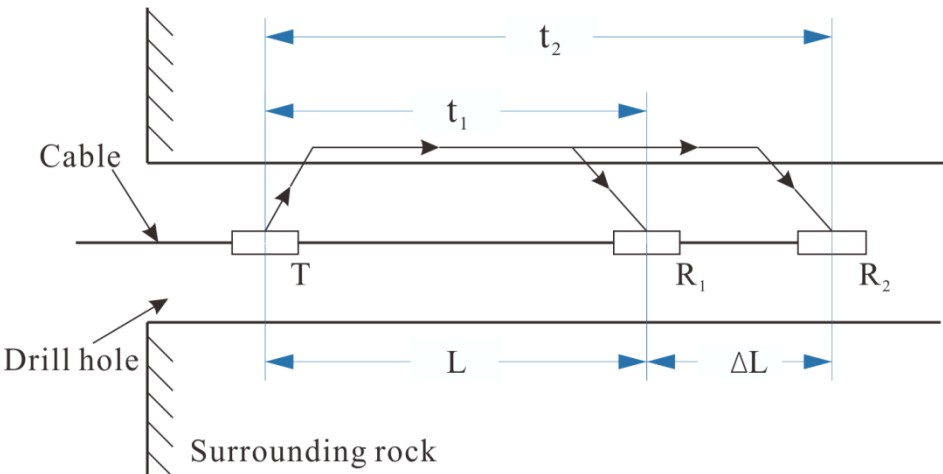

**Figure 9.** Operation principle of the single-emission and double-receiving measurement method.

The rock mass integrity is characterized by the integrity coefficient $K_v$ of the rock mass, which is calculated as follows:

$$K_v = (\frac{V_{p1}}{V_{p2}})^2,$$ (2)

where $V_{p1}$ is the rock mass's longitudinal wave velocity and $V_{p2}$ is the intact rock's longitudinal wave velocity. The intact marble's longitudinal wave velocity in the test roadway was approximately 6.5 km/s.

Based on the national standard, which classifies the rock mass in engineering (GB50218-94), the relationship between the rock mass integrity indicators $K_v$ and $V_p$ is shown in Table 3.

**Table 3.** Relationship between the rock mass integrity, $K_v$ and $V_p$.

| Rock Mass Integrity | Excellent | Good | Medium | Poor | Broken |
|---|---|---|---|---|---|
| $K_v$ | >0.75 | 0.75–0.55 | 0.55–0.35 | 0.35–0.15 | <0.15 |
| $V_p$ (km/s) | >5630 | 5630–4820 | 4820–3845 | 3845–2517 | <2517 |

Field measurements indicated that the characteristics of the four sections were similar and the results in Section 1 are shown in Figure 10. The rock mass below a medium integrity was considered to be the broken zone in this paper (that is, $K_v < 0.55$), and the broken rock zone in Section 1 is shown in Figure 11.

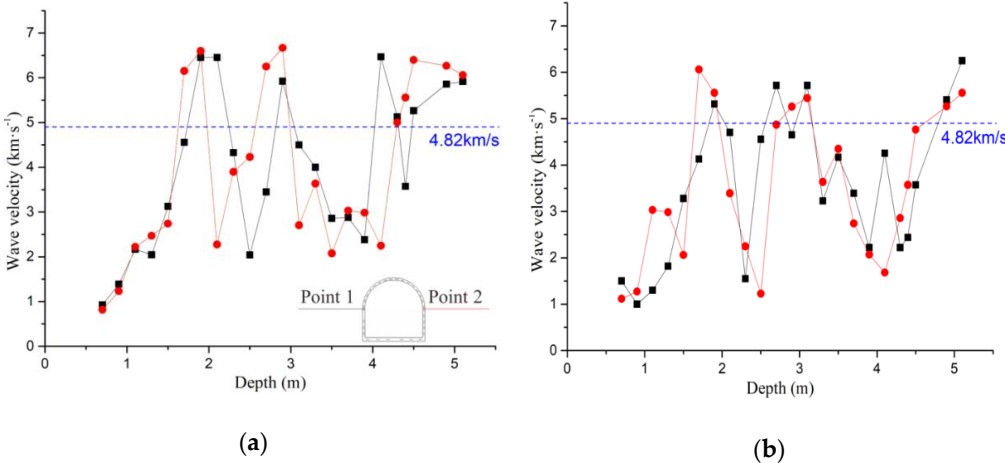

**Figure 10.** Longitudinal wave velocity curves in Section 1. (**a**) Measurement points on the spring line. (**b**) Measurement points on the spandrel.

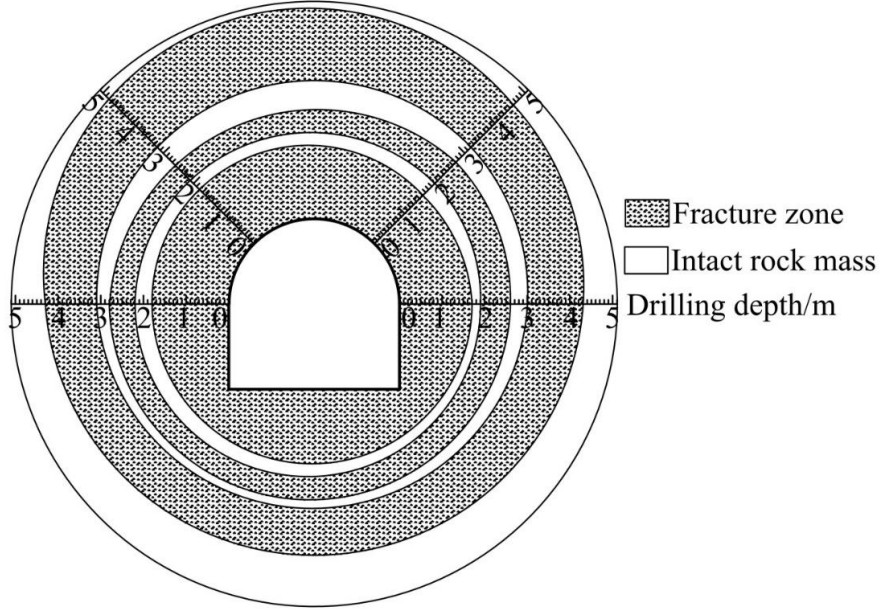

**Figure 11.** The zonal disintegration monitored in Section 1.

The figures above show that zonal disintegration phenomena occurred in the test roadway, which could be divided into three broken zones. The first one was within the range of 0–1.8 m from the roadway surface, with the largest scope and the most serious surrounding rock deformation. It exhibited a close relationship with roof falling and collapsing during the process of roadway excavation. This zone can be considered the broken rock zone in the traditional sense. A depth of about 2–2.8 m represents the second zone. This had a similar length as the 2.25 m bolt, which can explain the bolt failures found in the field survey. The third broken zone appeared at a distance of 3–4.8 m from the surface, and the zone on the spandrels was the largest, which was consistent with the frequent cracking on the spandrels. As the deepest borehole in this test was only 5.1 m, it is not known whether there was another zone with a distance greater than 5.1 m from the roadway. However, the discovery

of zonal disintegration was enough to prove that the anchoring method used in the original support was not reasonable and the bolt could not fully play its role, which needed to be further optimized.

## 4. "Multistage Anchorage System + CFST" Combined Support Technology

### 4.1. Establishment of the New Support Method

According to the above analysis, the disadvantages of the initial support are as follows:

1.  The U-shaped steel strength is insufficient under broken rock and high ground stress, as the steel tended to bend and even began to break, especially on the straight wall foot.
2.  Owing to the large fractured zone and zonal disintegration, the surrounding rock was not pierced completely for coupling as the bolts were not sufficiently long, which resulted in a low anchorage force and easy failure.
3.  There is no feasible measure to handle the floor deformation where the roadway deformation occurs and gradually develops.

Therefore, to make up for the defects of the initial support method, a "multistage anchorage system + CFST" joint support scheme is presented herein. The compound supporting structure includes a multistage anchorage system (bolt, cable, mesh, shotcrete) and, later, a high-strength rigid supporting structure (CFST). The design of the new support is displayed in Figure 12.

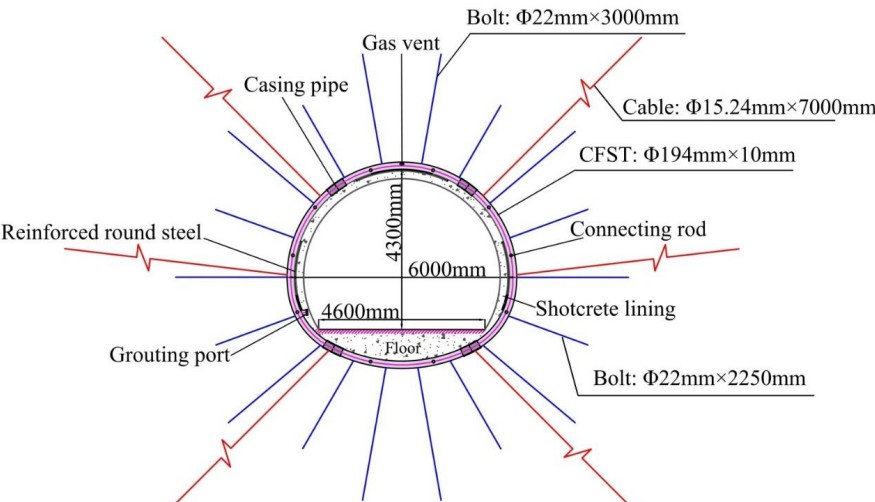

**Figure 12.** New support scheme layout.

4.1.1. Multistage Anchorage System

The multistage anchorage system in the new support scheme includes short bolts, long bolts, cables and their correlated wire mesh and shotcrete, as follows:

1.  Short bolts: 2.25 m full-length mortar bolts, with spacing of 1 m × 2 m, which are applied immediately after the roadway excavation and equipped with a 10 cm concrete spraying layer. The short bolts pass through the first complete rock mass layer, fixing the outermost broken zone to prevent roof falling and collapse during the construction process. This kind of flexible support can also release the original rock stress to some extent, diminishing the pressure for the forthcoming rigid support.
2.  Long bolts: 3 m full-length mortar bolts, with spacing of 1 m × 2 m. On the basis of the results of the broken rock zone test, the long bolts are able to pass through two broken zones and one intact zone to reach the second intact zone and fix this part of the rock together to completely exploit the rock mass's self-bearing capacity.

3.   Cables: 7 m, installed on the haunches, spandrels and corners on which large deformations frequently occur, to fully regulate the horizontal deformation of the roadways. Moreover, cables are connected with the surrounding rock reinforced by shallow combined bolts, forming a support system with a strong anti-deformation ability and greatly improving the stability and integrity of the surrounding rock.

Shallow broken surrounding rocks have a poor stress transfer ability. Through the multistage anchoring system, effective superposition of the surrounding rock stress is achieved and the synergistic effect is depicted in Figure 13. The multistage anchoring system combines the roadway surrounding rocks to form a reliable bearing structure. It also fully mobilizes the capacity of deep, stable surrounding rocks and restrains the development of harmful deformation. Engineering practice of the study area is taken into account when the length and raw materials of the bolts and cable are selected. For example, the anchor rod used in the study area is made of spiral steel with a 9 m standard length. Therefore, the bolt length (2.25 and 3 m) can effectively reduce the cost because there is no material waste.

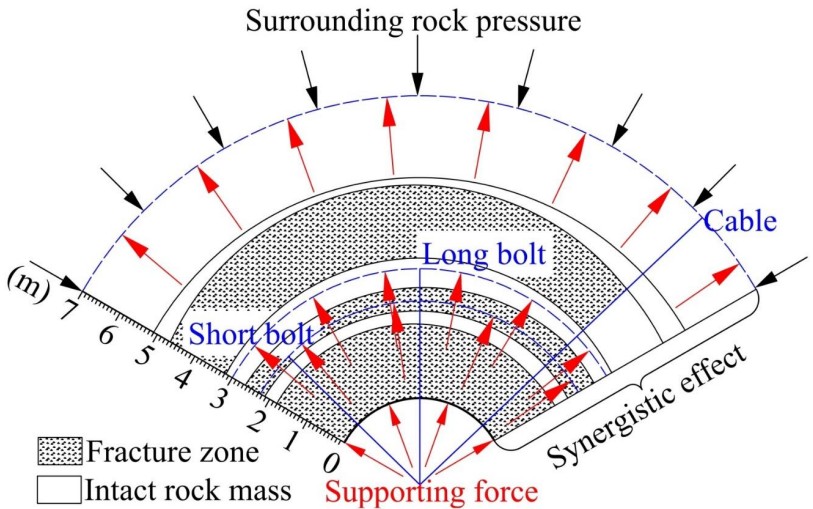

**Figure 13.** Synergistic effect of the multistage anchorage system.

### 4.1.2. CFST

CFST consisting of steel pipes and filled concrete is a passive support structure used to control roadway stability, especially in the case of terrible geological conditions, such as soft rock mass and high ground stress. The regional stability of the thin-walled steel tube is maintained by the inner concrete. In turn, the tube has a confining effect on the concrete, causing a tri-direction stress state and enhancing its strength. Therefore, the CFST has a high bearing capacity and can better control the surrounding rock mass's deformation.

The short column's ultimate bearing capacity ($N_0$) of CFSTs under axial compression can be measured using the following formula [17]:

$$\begin{cases} N_0 = A_c f_c (1 + \sqrt{\theta} + 1.1\theta) \\ \theta = A_s f_s / (A_c f_c) \end{cases}, \tag{3}$$

where $A_c$ is the cross-sectional area of the concrete (23,778 mm$^2$), $f_c$ is the strength of the concrete (25 N/mm$^2$), $\theta$ is the steel coefficient of the casing hoop, $A_s$ is the cross-sectional area of the tube (5780 mm$^2$) and $f_s$ is the steel material yield limit of 215 N/mm$^2$. Taking a reduction factor of 0.78 into account, the $N_0$ value obtained was 2012 kN.

Moreover, as illustrated in Figure 14, the supporting force ($\sigma$) can be calculated by:

$$s \int_{\pi/4}^{\pi/2} \sin\theta \sigma R d_\theta = N_0 \cos\frac{\pi}{4}, \tag{4}$$

where s is the steel tubes' distance and the tube radius is *R*.

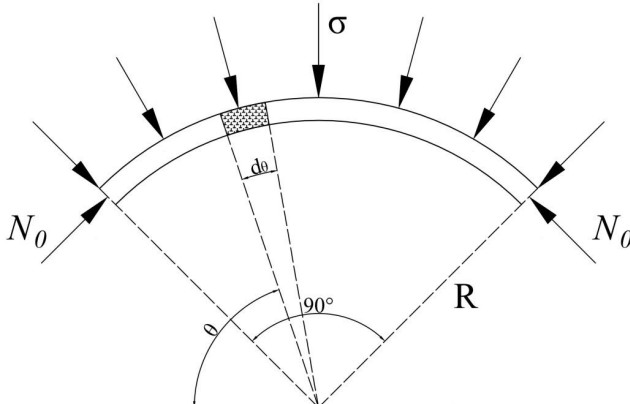

**Figure 14.** Mechanical model of a concrete-filled steel tube (CFST) arched support force schematic.

Through substitution, we obtained a result of 0.76 MPa for σ, which is greater than the 0.45 MPa supporting force of the U-shaped steel. Therefore, CFST is a better choice in this study area and makes up for the low strength of the U-shaped steel.

After CFST construction, supporting structures and the surrounding rock mass formed a hierarchical support system that cooperated and worked together to deal with roadway deformation, that is, the spray layer and the CFST formed the first level, which provided a normal constrained force for the surrounding rock on the face of the roadways. The bolt's second level improved the shallow broken rock mass's performance and the third level was formed of cables, which developed a continuous bearing structure through effectively connecting the shallow fractured rock and the deep intact rock.

*4.2. Numerical Simulations*

The numerical simulation test was conducted using FLAC$^{3D}$, so as to verify the new support method's supporting effect. As this software is good at simulating large deformation and the progressive failure process, it was selected herein.

4.2.1. Constitutive Models

Yang et al. [22] suggested that the surrounding rock's strength parameters will be noticeably nonlinearly decreased in an excavation. This is due to the micro-crack's constant expansion in the rock (i.e., strain-softening). When the peak strength is reached, the deterioration phenomenon will be more obvious when the deformation is still increasing. Yan et al. [5] and Yang et al. [11] found that the widely used material parameters in the Mohr–Coulomb model are preset constants, failing to show the surrounding rock's softening effect due to the variation of surrounding rock parameters. The failure criterion of Mohr–Coulomb, as well as the element's stress–strain relation in the strain-softening model, are illustrated in Figures 15 and 16. Following Zhang et al. [19], in the elastic phase, it can be observed that the element strain is merely elastic strain (ε_e), that is,

$$\varepsilon = \varepsilon_e. \tag{5}$$

The element strain comprises elastic strain ($\varepsilon_e$) and plastic strain ($\varepsilon_p$) when it is going to yield, that is,

$$\varepsilon = \varepsilon_e + \varepsilon_p. \tag{6}$$

The element yield function is,

$$\begin{cases} f^s = \sigma_1 - \sigma_3 N_\varphi + 2c\sqrt{N_\varphi} \\ f^t = \sigma_3 - \sigma^t \end{cases}, \tag{7}$$

where $f^s = 0$ is the shear failure, $f^t = 0$ is the tensile failure, $\sigma_1, \sigma_2$ is the maximum and minimum principal stress, respectively, $\sigma^t$ is the tensile strength, C is the cohesion, $\varphi$ is the internal friction angle and $N_\varphi = (1 + \sin\varphi)/(1 - \sin\varphi)$.

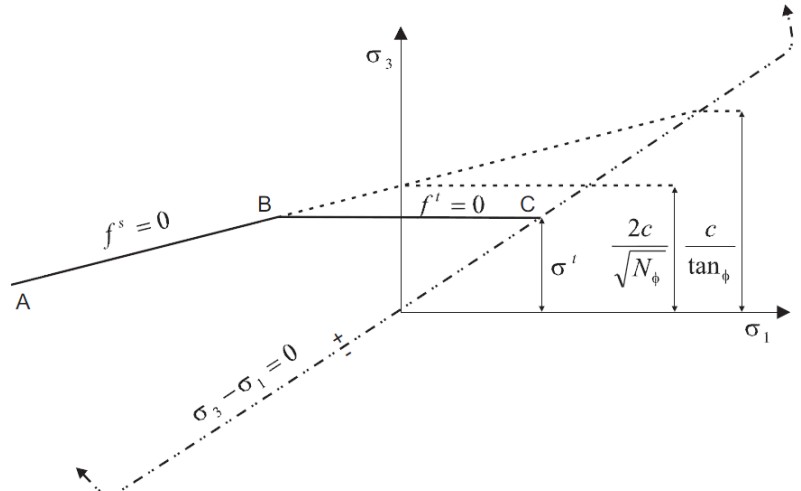

**Figure 15.** Failure criterion of Mohr–Coulomb.

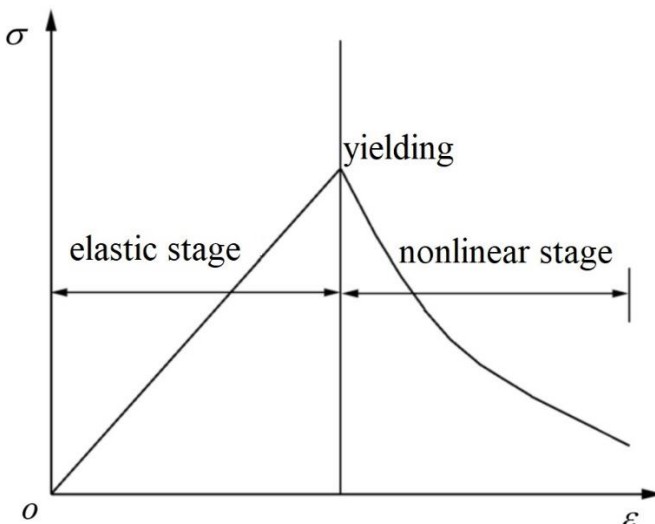

**Figure 16.** Element's stress–strain relation.

Therefore, the strain-softening model is used herein to depict the weak rock's nonlinear mechanical properties around the roadway.

### 4.2.2. Numerical Models

On the basis of the test roadway's practical standard size, a numerical model is constructed, as shown in Figure 17. The model dimension was 60 m × 20 m × 60 m (length × width × height). There were 199,080 grid points and 192,000 zones in the model. We set the displacement boundary in six different directions to regulate the normal displacement and applied the triaxial ground stresses in the model based on Table 2. Table 1 shows the marble rock mass's mechanical parameters in the test roadway. On the basis of the software's structural unit, the support structures were simulated: the beam unit was simulated for CFST, the cable unit was simulated for the bolt and cable, and the shell unit was simulated for the concrete spray layer. Each support structure's detailed parameters are referenced in Li et al. [33].

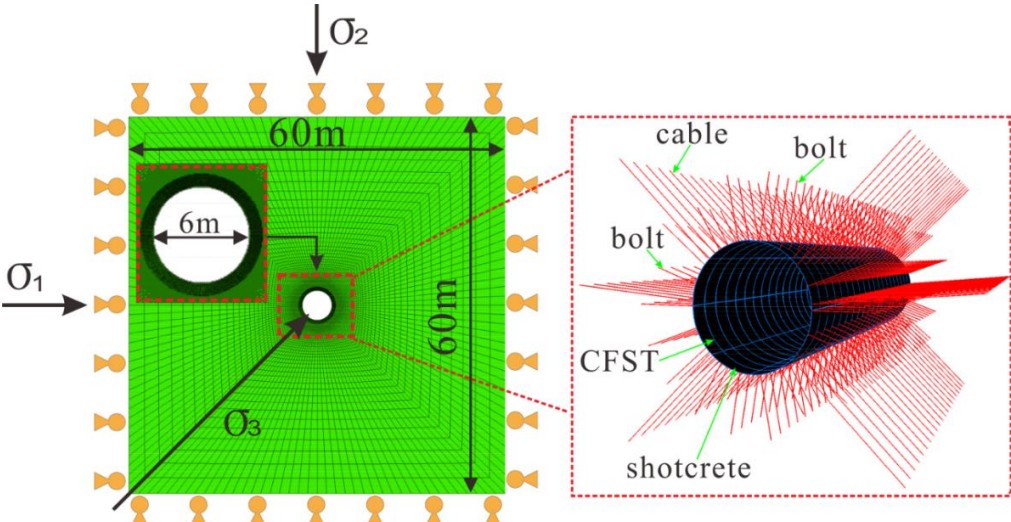

**Figure 17.** Numerical model of the initial support.

### 4.2.3. Simulation Results

In order to highlight the effect of the new support scheme, a comparative test was carried out to simulate the deformation of a roadway without support.

When a roadway is excavated without support, the simulation calculation cannot converge or balance, and the surrounding rock deformation will continue. Figures 18 and 19 show the results of the 5000 step calculation. It can be seen that the deformation of a roadway surrounding rocks is large, with the deformation on the two sides exceeding 70 cm and the deformation on the top and bottom exceeding 60 cm. A wide range of surrounding rock is affected by the roadway excavation and the distribution of the plastic zone is roughly an ellipse, with a long axis radius of 15 m and a short axis radius of 12 m. The direction of the long axis is nearly parallel to the Z axis, which is closely related to the great horizontal ground stress. The surrounding rocks in the plastic zone all experienced a shear yield state, and the surrounding rocks in the depth range of 2 m from the roadway surface experienced a tensile shear state, which is very unfavorable for the stability of the roadway.

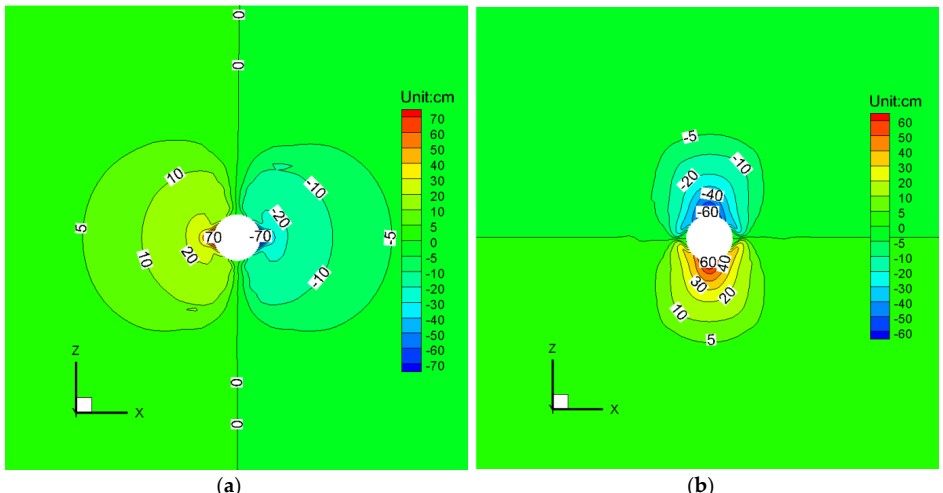

**Figure 18.** The surrounding rock mass's displacement without the support scheme: (**a**) Horizontal displacement., (**b**) Vertical displacement.

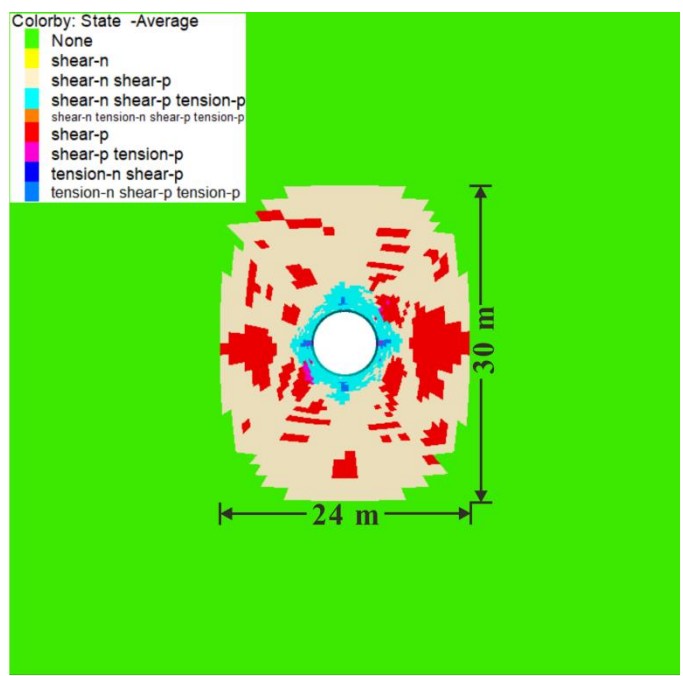

**Figure 19.** The surrounding rock's plastic zone distribution without the support scheme.

Figure 20 depicts the simulation results of the surrounding rock mass's horizontal and vertical displacement under the new support scheme.

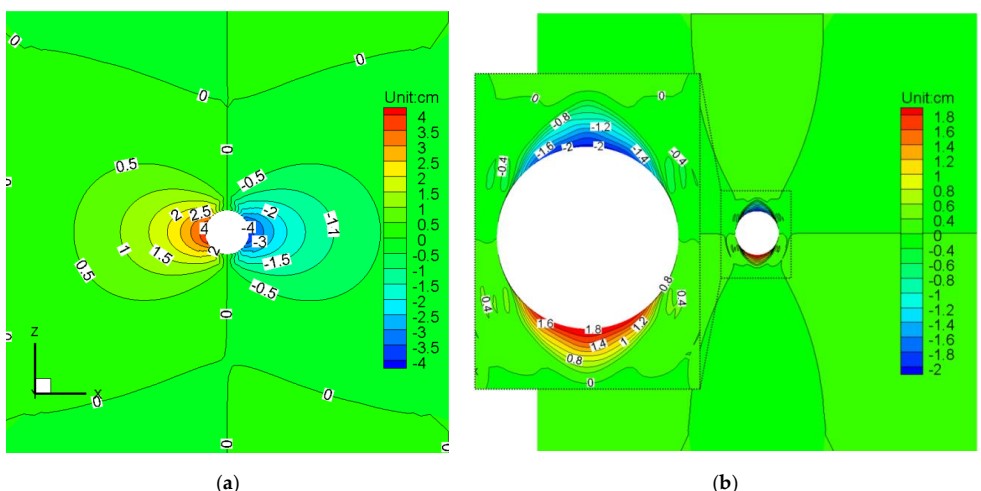

(**a**)　　　　　　　　　　　　　(**b**)

**Figure 20.** The surrounding rock mass's displacement under the new support scheme: (**a**) Horizontal displacement, (**b**) Vertical displacement.

It can be seen that the new support method can effectively control the roadway's convergence deformation. Especially in the vertical direction, the vault and floor's maximum deformation is only about 2 and 1.8 cm, respectively, and the scope of the deformation influence is rather small. Meanwhile, a 4 cm cumulative deformation appears on both sides, which is also an acceptable deformation in a deep roadway.

Figure 21 displays the surrounding rock's plastic zone distribution. Around the roadway, the plastic zone has a uniform distribution, and the maximum influence distance is approximately 2 m on the two sides and 3 m on the vault and floor. Moreover, tension shear failure only took place on the roadway surface and nearly became stable. All of the numerical simulation results suggest the effectiveness of the new support scheme's control effect.

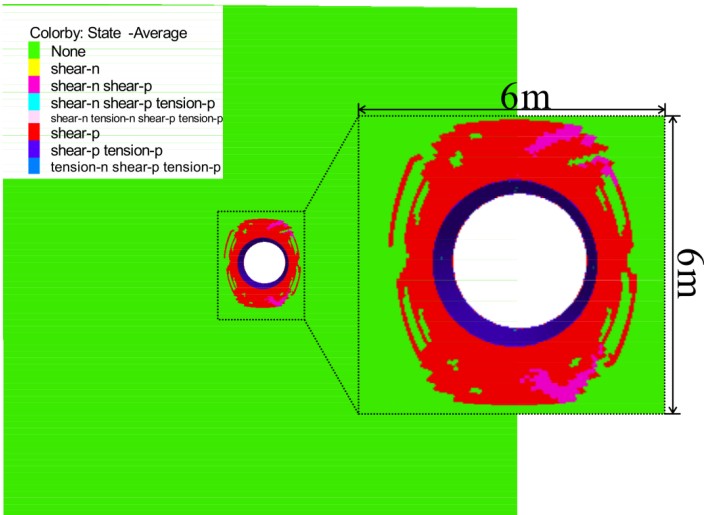

**Figure 21.** The surrounding rock's plastic zone distribution under the new support scheme.

### 4.3. Field Monitoring

Except for numerical simulations, the surface convergence of the tentative roadway and the deformation of the CFST was monitored after the new support method had been applied. We chose five convergence monitoring sections (spacing = 10 m) in the tentative roadway and selected six monitoring points for each section. Only horizontal convergence was measured because the previous analysis showed that horizontal deformation was serious in the study area. Moreover, six CFSTs were selected, and four steel string strain gages were installed in each of these. The measurements lasted nearly 500 days and more than 20 groups of data were obtained.

The average displacement curves of the surrounding rocks after maintenance are shown in Figure 22a, from which it is possible to see that the roadway shrank rapidly in the first 80 days and deformation then attenuated, after which it tended to become stable after about 320 days. At the last measurement, the two sides moved close to 75 to 90 mm, which is in agreement with the results of the numerical simulation. Moreover, Figure 22b reveals that the displacement of the CSFTs increased with time and the maximum deformation occurring on the left was only about 28 mm, which indicates that the tubes supply a support force and remain stable after more than 400 days.

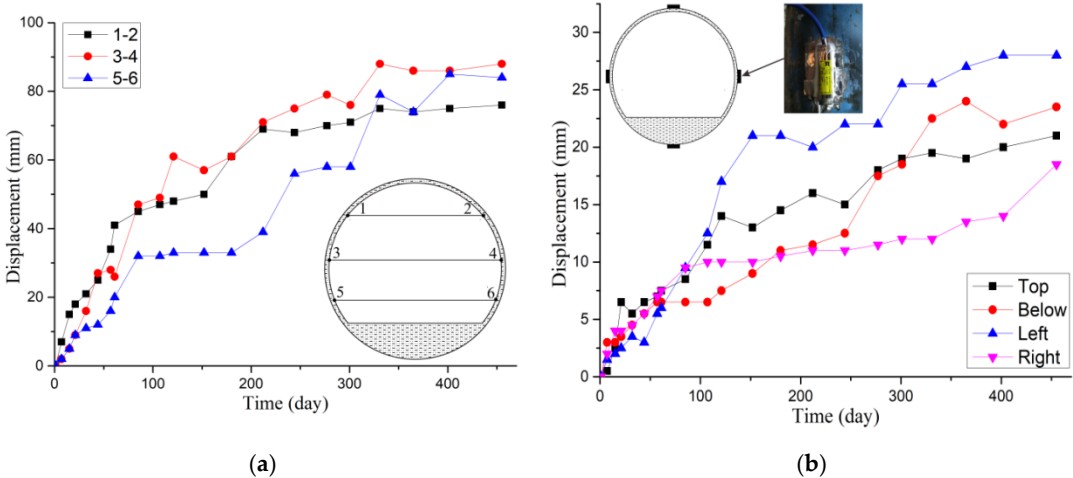

(**a**)　　　　　　　　　　(**b**)

**Figure 22.** Results of the site monitoring: (**a**) Curves of convergent deformation, (**b**) Displacement curves of CFST.

The results of the numerical simulation and field monitoring and the photographs of the tentative roadway before and after the novel support scheme (depicted in Figure 23), show that the improved support "multistage anchorage + CFST" is able to control the surrounding rock's deformation in the deep roadway under high ground stress and broken rock mass and ensure that the roadway is stable.

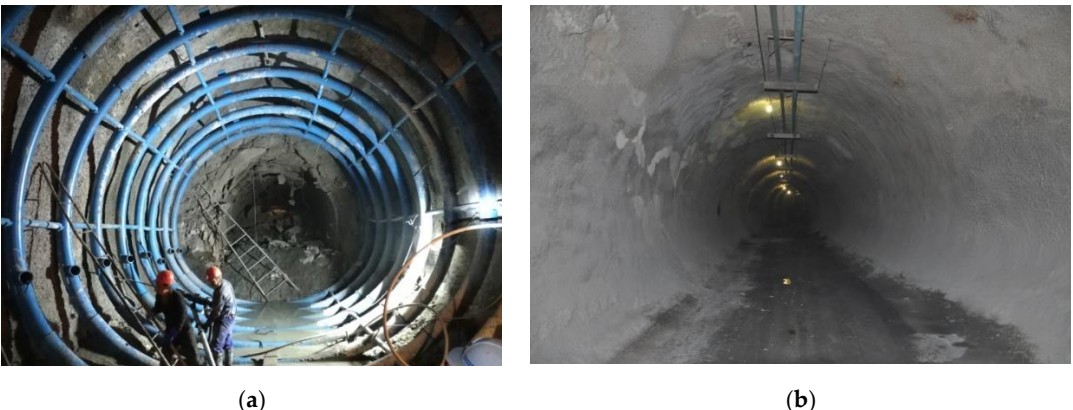

| (**a**) | (**b**) |

**Figure 23.** Schematic of the test roadway before and after the novel support scheme: (**a**) Construction period, (**b**) Current situation.

## 5. Conclusions

This paper has discussed a case study on the characteristics of deformation and the control technology of deep roadways under broken high ground stress and rock mass in Jinchuan No. 2 Mine. On the basis of a detailed field investigation, the deformation failure types and characteristics in the roadway under primary support have been outlined. Additionally, a numerical simulation and field monitoring were performed to assess the influence of the improved support in the test roadway. The main conclusions of the research are as follows:

(1) The roadway deformation failure types could be sorted into six types, including roof cracking, floor heaving, unsymmetrical pressure, roof falling, side cracking and rib spalling, among which rib spalling is the most severe. Moreover, the main features of roadway deformation include prominent time effects and significant space effects, and the maximum principal stress plays a dominant role.

(2) The broken rock zone test found that zonal disintegration phenomena occur in the test roadway and there are three broken rock zones whose depths are approximately 0–1.8, 2–2.8, and 3–4.8 m. The zonal disintegration proves that the anchoring method used in the original support is not feasible.

(3) An improved combined supporting scheme called the "multistage anchorage + CFST" has been put forward and assessed. The supporting structures and surrounding rock mass formed a hierarchical support system which could cooperate and work together. The results of the numerical simulation and field monitoring indicate the effectiveness of the new support scheme's control effect.

**Author Contributions:** Data curation, G.L.; Formal analysis, G.L., H.Z. and F.M.; Methodology, G.L., and F.M.; Software, G.L. and J.G.; Writing—original draft, G.L. and J.G. All authors have read and agreed to the published version of the manuscript.

**Funding:** The research was funded by the National Natural Science Foundation of China (Grant Nos. 41831293, 41877274, and 41772341). Grateful appreciation is expressed for the support.

**Acknowledgments:** The authors would like to express their sincere gratitude to Jinchuan mine for their data supports. In addition, the authors are grateful to assigned editor and three anonymous reviewers for their enthusiastic help and valuable comments which have greatly improved this paper.

**Conflicts of Interest:** The authors declare no conflict of interest.

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
