# Peer review of "Deformation Characteristics and Control Method of Kilometer-Depth Roadways in a Nickel Mine: A Case Study"

_applsci, doi:10.3390/app10113937_

Round 1
Reviewer 1 Report
The work brings a new control method of multistage anchorage+concrete-filled steel tube in kilometer-depth roadways in metallic mines under high ground stress.
The article has an imminent nature of applied science, with relevance and novelty for ​​geomechanics area.
The article is well structured, has a guiding thread that facilitates its reading, although the section titles (e.g. "Geological environment and engineering properties") do not facilitate their connection to the content.
The introduction is very well written. However, between line 47 and 373, the text needs a careful Englis review.
It is pointed out below some minor aspects that need to be corrected:
line 43 - "the study area in this study" - consider to correct the repetition;
line 64 - "monoclinic" - does the authors mean "monocline"?
line 169 - "mechanical mechanism" - consider to use other word for mechanism;
line 357 - "analyze" - does the authors mean "analysis"?
line 367 - "results, results" - consider to correct the repetition;
equations 5 and 7 - correct line wrapping in parentheses;
Figure 18 a) and b) consider to correct the color bar that is cut;
Figure 20 a) and b) these figures are also cut.
Author Response
First of all, the authors would like to express their sincere gratitude to you for the valuable comments which have greatly improved this paper. The revisions are clearly highlighted by the “Track Changes” function in Microsoft Word and the explanations of your comments point-by-point are shown as followings:
Point 1: line 43 - "the study area in this study" - consider to correct the repetition;
Response 1: One of the “study” has been replaced by “research” in the revision.
Point 2: line 64 - "monoclinic" - does the authors mean "monocline"?
Response 2: Yes, it is really a spelling mistake and modified in the revision.
Point 3: line 169 - "mechanical mechanism" - consider to use other word for mechanism;
Response 3: “mechanism” has been replaced by “principle”.
Point 4: line 357 - "analyze" - does the authors mean "analysis"?
Response 4: A noun “analysis” should be used here, and it has been modified in the revision.
Point 5: line 367 - "results, results" - consider to correct the repetition;
Response 5: It has been modified in the revision.
Point 6: equations 5 and 7 - correct line wrapping in parentheses;
Response 6: The equations in the revision have been renumbered
Point 7: Figure 18 a) and b) consider to correct the color bar that is cut;
Response 7: It has been improved in the revision.
Point 8: Figure 20 a) and b) these figures are also cut.
Response 8: It has been improved in the revision.
Besides, the grammar, spelling, punctuation etc. are improved by the English editing service provided by MDPI.

Reviewer 2 Report
This is a very interesting case study but it needs many improvements prior to publication. These are presented in detail below:
- Linguistic: the language of the paper is average and should definitely be improved grammatically and spelling-wise prior to consideration for publication. Also, several words or adjectives that indicate emotional or verbal communication methods should be replaced by more neutral and scientific ones. For example, words such as "hot research", "hugest ore body", "terrible geological conditions", "detailedly", "occurrence times", "unloading is fierce", etc., should be replaced by more scientific terms.
- Editorial: reference to Figures 3 and 4 should be switched around. Where it says "depicted in Figure 3", figure 4 should be referred to and where it says "as shown in Figure 4", it is figure 3 that should be referred to.
- Technical: several technical modifications are needed as the current paper raises some questions that need clarification:
- It is stated that stresses are very high in this region but the maximum principal stress does not exceed 50 MPa. This cannot be considered as very high for 1000 m depth. Furthermore, since the rockmass is heavily fractured and soft (line 144), it is strange to have even the current values in such a setting. Stresses of 25 MPa (vertical) and 50 MPa (horizontal) would be normal for 1000 m depth but with fractured rockmass, these values should be even lower.
- More information should be provided for the broken rock test as to when it is used and what does it achieve. Some description is given in lines 188-205 but this is more about its procedure in the field. A background review should be made as to its usage in mining, and how its classification (Table 3) compares to rockmass classification systems such as RMR or Q.
- The use of short bolts in the multistage system is not fully explained. If the rockmass around the excavation is fractured, where do the short bolts anchor the broken parts to? In figure 13, there seems to be an intermediate intact rockmass region where this anchoring is established but this is not mentioned in the text in lines 246-251.
- The most puzzling technical issue is the numerical simulation in FLAC3D. In figure 4 the tunnel is shown as a horseshoe-shaped one, in figure 12 it is depicted as being ovaloid in shape, and in figure 17 it is a circle. Each of these shapes has certain impacts on the stress redistribution and cannot be compared to one another. Which is the shape of the actual tunnel at the mine?
- FLAC3D is a very difficult code to work with and the authors have been able to use it, which is commendable. However, they have not taken advantage of its 3D capabilities and the model is only a 2D one that has been extended in the 3rd dimension. The reader does not see the side tunnels shown in figure 3, which is a pity since the model can handle the real geometry of tunnel networks at the mine.
- All models need to be calibrated before using them for prediction purposes. Calibration is done by comparing the model output to rockmass behaviour in the field. The authors have not done this and do not show the model showing failure and squeezing before implementing the support system. There should be two FLAC3D models; in the first no support/basic support should be used and it should indicate the rockmass failures in the field. In the second, and with the installation of the multistage support, the model should show minimal squeezing and plastic zone disturbance as in figure 19. This is when it can be said that the model is fully calibrated and validated.
- A practical shortcoming of the analysis and proposal of the new support system is that it works with the current tunnel system at the mine. It means that all the expensive multistage system can be used to keep the tunnel stable in the current conditions. No mention is made as to what will happen when mining takes place or what mining method will be used. It is very well known that opening up stopes of much larger dimensions near the tunnels will only increase the stress on the support system, which might not be able to bear it. The authors should address this issue so that the study is of practical use. It is not an achievement to put up a complex system that can only hold up the tunnel system when no mining is taking place.
The paper covers a very good case study mine and has very good potential. Once these technical, editorial, and linguistic issues are dealt with, it should be resubmitted for review/publication.
Author Response
First of all, the authors would like to express their sincere gratitude to you for the valuable comments which have greatly improved this paper. The revisions are clearly highlighted by the “Track Changes” function in Microsoft Word and the explanations of your comments point-by-point are shown as followings:
Linguistic: the language of the paper is average and should definitely be improved grammatically and spelling-wise prior to consideration for publication. Also, several words or adjectives that indicate emotional or verbal communication methods should be replaced by more neutral and scientific ones. For example, words such as "hot research", "hugest ore body", "terrible geological conditions", "detailedly", "occurrence times", "unloading is fierce", etc., should be replaced by more scientific terms.
Response: The grammar, spelling, punctuation etc. are improved by the English editing service provided by MDPI. Thanks for your careful review and the deficiencies you mentioned are improved in the revision.
Editorial: reference to Figures 3 and 4 should be switched around. Where it says "depicted in Figure 3", figure 4 should be referred to and where it says "as shown in Figure 4", it is figure 3 that should be referred to.
Response: It was really a mistake when I organized the paper and has been modified in the revision.
Technical: several technical modifications are needed as the current paper raises some questions that need clarification:
Point 1: It is stated that stresses are very high in this region but the maximum principal stress does not exceed 50 MPa. This cannot be considered as very high for 1000 m depth. Furthermore, since the rockmass is heavily fractured and soft (line 144), it is strange to have even the current values in such a setting. Stresses of 25 MPa (vertical) and 50 MPa (horizontal) would be normal for 1000 m depth but with fractured rockmass, these values should be even lower.
Response 1: We have conducted a long time of research in Jinchuan area, and the parameters given in this paper are all obtained from field measurement. There are also relevant papers that can provide evidence for the data in this paper, such as:
1. Study on deformation failure mechanism and support technology of deep soft rock roadway, Engineering Geology, 2020: 264, 105262.
2. Field investigations of high stress soft surrounding rocks and deformation control. Journal of Rock Mechanics and Geotechnical Engineering, 2015, 421e433.
3. Ground movement resulting from underground backfill mining in a nickel mine (Gansu Province, China). Nat Hazards. 2015: 77:1475-1490.
Therefore, there is no problem with the data in this paper. If you think that the description of “very high ground stress” is not accurate enough, I can make corresponding changes.
Point 2: More information should be provided for the broken rock test as to when it is used and what does it achieve. Some description is given in lines 188-205 but this is more about its procedure in the field. A background review should be made as to its usage in mining, and how its classification (Table 3) compares to rockmass classification systems such as RMR or Q.
Response 2: The parameters of several kinds of rock around the test roadway are given in Table 1, so that readers can understand the engineering geological characteristics of “high intact rock strength and low rock mass strength” in the study area. This study is mainly an engineering scale study without much description of the rock test and parameter acquisition. The data given in Table 3 are the results of acoustic detection. According to Figure 4, the surrounding rock of the test roadway is marble, and the RMR of this rock is provided in Table 1. However, one of the two results is the in-situ test, and the other one is an overall assessment, so the comparison is of little significance.
Point 3: The use of short bolts in the multistage system is not fully explained. If the rockmass around the excavation is fractured, where do the short bolts anchor the broken parts to? In figure 13, there seems to be an intermediate intact rockmass region where this anchoring is established but this is not mentioned in the text in lines 246-251.
Response 3: It can be seen from the loose circle test results that the roadway with a buried depth of 1000 m in the study area shows obvious zonal disintegration. The anchorage zone acted by the short bolt is the outermost complete zone. Of course, the situation in other regions may be different, and the short bolt of 2.25m is suitable for this study area. This is also revealed in the manuscript: “The short bolts passes through the first complete rock mass layer, fixing the outermost broken zone to prevent the roof fallings and collapses during the construction process. This kind of flexible support can also release the original rock stress to some extent, diminishing the pressure for the forthcoming rigid support.”
Point 4: The most puzzling technical issue is the numerical simulation in FLAC3D. In figure 4 the tunnel is shown as a horseshoe-shaped one, in figure 12 it is depicted as being ovaloid in shape, and in figure 17 it is a circle. Each of these shapes has certain impacts on the stress redistribution and cannot be compared to one another. Which is the shape of the actual tunnel at the mine?
Response 4: When the roadway in the study area is initially excavated, the section shape is horseshoe as shown in Figure 3. Figure 12 shows the section shape of concrete-filled steel tube support, which is an oval. When concrete-filled steel tube support is used for roadway repair, the roadway needs to be excavated into a circular section. During the simulation calculation in FLAC3D, the roadway section is generalized into a circle. The roadways introduced in the study are all based on the actual conditions of mining and repair.
Point 5: FLAC3D is a very difficult code to work with and the authors have been able to use it, which is commendable. However, they have not taken advantage of its 3D capabilities and the model is only a 2D one that has been extended in the 3rd dimension. The reader does not see the side tunnels shown in figure 3, which is a pity since the model can handle the real geometry of tunnel networks at the mine.
Response 5: The models established in this paper can be regarded as a two-dimensional model and only a roadway cross-section is studied. However, as you know, FLAC3D is a very difficult code, and it is not easy to build a 3D model considering all actual conditions of the mine. The author is also trying to optimize the numerical model, hoping to obtain more valuable conclusions in the future research.
Point 6: All models need to be calibrated before using them for prediction purposes. Calibration is done by comparing the model output to rockmass behaviour in the field. The authors have not done this and do not show the model showing failure and squeezing before implementing the support system. There should be two FLAC3D models; in the first no support/basic support should be used and it should indicate the rockmass failures in the field. In the second, and with the installation of the multistage support, the model should show minimal squeezing and plastic zone disturbance as in figure 19. This is when it can be said that the model is fully calibrated and validated.
Response 6: According to your opinion, a comparative test of no support is added in the revision, which increases the credibility of the numerical simulation results. Thank you for your suggestions for improvement.
Point 7: A practical shortcoming of the analysis and proposal of the new support system is that it works with the current tunnel system at the mine. It means that all the expensive multistage system can be used to keep the tunnel stable in the current conditions. No mention is made as to what will happen when mining takes place or what mining method will be used. It is very well known that opening up stopes of much larger dimensions near the tunnels will only increase the stress on the support system, which might not be able to bear it. The authors should address this issue so that the study is of practical use. It is not an achievement to put up a complex system that can only hold up the tunnel system when no mining is taking place.
Response 7: According to your opinion, the mining method and present situation of the mining area are added in the Chapter 2. The ore body with a depth of 1000 m in the mining area has not been exploited, and the construction of foundation engineering is carried out near the test roadway. It is difficult to accurately describe the mining pressure, and we will make further analysis on the later field monitoring data. Moreover, what needs illustration is that the roadway studied in this paper is a transport roadway in the mining area, which belongs to a permanent engineering, so it is of great significance to ensure its safety and stability.

Round 2
Reviewer 2 Report
Most of the comments and suggestions provided have been implemented in the revised version, and the language of the text has been improved. While some of the recommendations have not been followed, the paper has undergone sufficient improvement to merit publication.